# A Pilot Nurse-Administered CBT Intervention for Insomnia in Patients with Schizophrenic Disorder: A Randomized Clinical Effectiveness Trial

**DOI:** 10.3390/jcm12196147

**Published:** 2023-09-23

**Authors:** David Batalla-Martin, Maria-Antonia Martorell-Poveda, Angel Belzunegui-Eraso, Alejandro Marieges Gordo, Helena Batlle Lleal, Raquel Pasqual Melendez, Raquel Querol Girona, Marina López-Ruiz

**Affiliations:** 1Nou Barris Mental Health Center, 08016 Barcelona, Spain; alex.marieges@csm9b.com (A.M.G.); helena.batlle@csm9b.com (H.B.L.); raquel.pasqual@csm9b.com (R.P.M.); raquel.querol@csm9b.com (R.Q.G.); 2Nursing Department, Faculty of Nursing, Rovira and Virgili University, 43002 Tarragona, Spain; mariaantonia.martorell@urv.cat; 3Social Inclusion Chair, Rovira i Virgili University, 43002 Tarragona, Spain; angel.belzunegui@urv.cat; 4Service of Psychiatry and Psychology, HM-Sant Jordi Clinic, 08030 Barcelona, Spain; marinalopezr@gmail.com

**Keywords:** insomnia, schizophrenia, sleep disturbance, quality of life, cognitive behavioural therapy for insomnia

## Abstract

Insomnia is a highly prevalent disorder among the population with schizophrenia and has a significant impact on their quality of life. Cognitive behavioural therapies (CBT) have shown effectiveness in the treatment of insomnia in the general population. The aim of this this pilot study was to evaluate the effectiveness of a group intervention led by nurses in an outpatient mental health centre. The group work combined cognitive behavioural and psychoeducational therapeutic interventions to improve insomnia in patients with schizophrenic disorder and their health-related quality of life. This randomized clinical trial included intervention and control groups with follow-up assessments at 6 and 9 months, using the Insomnia Severity Index (ISI), Pittsburgh Sleep Quality Index (PSQI), and EuroQol-5D (EQ-5D) scales. The inclusion criteria were as follows: over 18 years of age, diagnosis of schizophrenia, and a score of >7 on the ISI scale. The total sample was 40 participants. The ISI scale showed a mean difference of 3.63 (CI 95%: 2.02–5.23) (*p* = 0.000) and 4.10 (CI 95%: 2.45–5.75) (*p* = 0.000) and a large effect size (F: 28.36; *p* = 0.000; ηp2: 0.427). Regarding the PSQI scale, the mean difference was 3.00 (CI 95%: 1.53–4.49) (*p* = 0.000) and 2.30 (CI 95%: 0.85–3.75) (*p* = 0.000), with a medium effect size (F: 18.31; *p* = 0.000 ηp2: 0.325). The EQ-VAS scale showed a difference in mean scores between the groups of 10.48 (CI 95%: −19.66–−1.29) (*p* = 0.027). CBT adapted for populations with mental disorders, carried out by nurses, is effective in improving insomnia and health-related quality of life.

## 1. Introduction

Sleep is an unconscious temporal physiological state characterized by a cessation of sensorial activity, mobility, and alertness. It consists of an active state where changes in corporal functions and mental activity of significant transcendence take place for psychological and physical balance [1].

Sleep disorders can represent an important risk for patients by producing somatic and psychological disorders and negatively affecting quality of life. It has been demonstrated that certain pathologies related to sleep can constitute an important risk factor for health, independent of aspects, such as age, sex, obesity, or tobacco consumption, among others [2]. Insomnia is the most present sleep alteration in psychiatry [3] and the most frequent disorder in the general population, affecting 40% of the population [4].

The prevalence of insomnia is identified, according to Hou et al. [5], in 28.9% of the population with schizophrenic disorder. Cohrs’ analysis in 2008 [6] confirmed that sleep disorders existed between 30% and 80% of individuals with schizophrenia. Kaufmann et al. [7] placed a prevalence of some symptoms of insomnia at 78%, and a study from Mondal et al. [8] affirmed that problems ascended to 83.4% of the sample. 

Insomnia’s impact on quality of life can be considerable [9]. Moreover, this disorder can affect many aspects of life, such as the ability to perform daily tasks and enjoy social activities [10]. Individuals who report insomnia have higher morbidity and mortality rates [10]. Furthermore, Benson et al. [11] explained that insomnia can be an important marker as a prodromal symptom of psychiatric decompensation or a exacerbation of symptomatology in patients with schizophrenic disorder. Patients who suffer from this disorder show an increased frequency of healthcare consults, which suggests an attendant rise in direct and indirect costs. Insomnia is still a poorly recognized, diagnosed, and treated condition [12,13].

The treatment for sleep disorders and insomnia is usually pharmacologic. However, evidence proves that cognitive behavioural and psychoeducational interventions have shown major efficacy in long-term treatment, prognosis and even in reducing economic costs [14]. It is also the therapy of choice for insomnia’s treatment according to the European Guideline for the Diagnosis and Treatment of Insomnia [15].

The characteristics of CBT consist of a therapeutic relationship of collaboration, structured sessions, and both cognitive and behavioural change techniques, focusing on the symptom and the disorder itself [16]. This technique has proven effective in various studies [17,18,19].

CBT has also shown statistically significant results in the treatment of insomnia [20,21] and in the treatment of patients with organic and mental health pathologies [22]. The CBT-I has also shown efficacy in patients with schizophrenic disorder and has been studied in group interventions [23], individuals [24,25], and also in interventions during hospital admission [26,27]. In addition, efficacy has been observed in brief CBT-I interventions in patients with depressive disorder [28].

The adaptation of interventions to the target population is an important aspect [29] for optimal effectiveness [30]. Thus, the study prepared by Waite et al. [31] describes different aspects to be considered in the adapted CBT-I interventions, for patients with schizophrenia, where it stands out: sleep disrupted by voices/paranoia, nightmares, fear of bed or, poor sleep environment. The findings observed by Batalla-Martin et al. [32], through the narrative of the subjects’ experience of insomnia and their strategies to manage it, allow adaptations based on: rumination prior to reconciliation; lack of daytime activity; knowledge about the effectiveness of CBT interventions (instead of pharmacological interventions). Also, health-related quality-of-life variables, with a greater affectation and influence on insomnia, in patients with a mental disorder are important for adaptations [33]. Other adaptations, such as the one made by Hwang et al. [23], aim to reduce or increase the number of sessions [24], reaching up to eight individual sessions, also carried out at home in a study by Freeman et al.

The aim of this pilot study is to evaluate CBT’s effectiveness in insomnia’s treatment and the improvement in quality of life and the maintenance of these advances during the posterior 6 and 9 months, in a sample of patients diagnosed with schizophrenic disorder and who maintain ambulatory follow-ups in the Nou Barris Mental Health Centre, Barcelona. The study’s design has followed guidelines and CONSORT’s considerations [34] and has been registered on Clinicaltrial.gov, accessed on 20 October 2021 (ID number NCT05115604).

## 2. Materials and Methods

A randomised clinical effectiveness trial was conducted between February 2021 and April 2022, in an outpatient Mental Health Centre in Barcelona, Spain.

### 2.1. Subjects

A total of 40 participants were recruited during the months of February and March of 2021 by the nursing team in the Nou Barris Mental Health Centre, Barcelona.

Inclusion criteria were as follows: (a) patients from 18 years of age of both sexes diagnosed with 295.xx Schizophrenia according to DSM-IV criteria or F20.xx Schizophrenia according to ICD-10; (b) patients who present a <7 score in the Insomnia Severity Index (ISI); and (c) patients must sign the informed consent and receive an informative document. The exclusion criteria were (a) presentation of terminal organic pathology, (b) lack of knowledge of the Catalan or Spanish language, (c) illiteracy, (d) diagnosis of mild or severe mental retardation, and (e) neurological disease involving cognitive impairment (e.g., dementia).

The study rigorously followed international ethical recommendations for medical research and was created under the ethical principles specified in the Declaration of Helsinki. The participants were also informed that all data would be treated as confidential in accordance with regulation (EU) no. 2016/679 and Spanish Organic Law 3/2018 of 5 December on the Protection of Personal Data and the guarantee of digital rights. The study received prior approval from the Ethics Committee of the Unió Catalana de Hospitals (UCH) with registration code CEI 21/13.

### 2.2. Procedure

#### 2.2.1. Sample Selection

The Nou Barris Mental Health Centre nursing team was in charge of identifying patients with schizophrenia (according to their clinical history) who presented an insomnia disorder in relation to the sample of the previous study [33]. In the previous study, 267 patients with schizophrenia were interviewed to determine the prevalence of insomnia evaluated with the Insomnia Severity Index (ISI) scale, with 110 of them presenting a score greater than 7 on the ISI scale. Once the total sample was selected, patients were invited to participate in the effectiveness study of the CBT-I intervention. Once they were informed of the study, the Insomnia Severity Index (ISI) was used again to select patients who would participate in the posterior study. Of the previous 110 patients, 63 agreed to participate, of which only 40 met the inclusion criteria for this pilot study. Patients with mild, moderate, or severe insomnia (ISI score > 7) who agreed to participate were provisionally accepted in the trial. Once they were selected, the patients were interviewed individually by their nurse, and the first data collection of the investigation protocol was conducted.

All patients following up at the mental health center with a diagnosis of severe mental disorder have visits with their referring psychiatrist and nurse, and no other CBT group interventions are currently offered at the center.

#### 2.2.2. First Evaluation and Randomisation

The initial evaluation included sociodemographic characteristics (age, sex, marital status, education level, employment status), medication (anxiolytic antidepressants, hypnotics), ISI scale administration, the Pittsburgh Sleep Quality Index (PSQI) questionnaire, and the EquoQol-5D Scale (EQ-5D). All the study’s variables were introduced in a database. The randomisation was conducted through the SPSS v21 programme.

#### 2.2.3. Evaluation Follow-Up

All of the resulting variables were evaluated three times: before the beginning of the study (pre-intervention) and at 6 and 9 months after inclusion (first and second follow-up, respectively) in individual sessions. Some reasons have been taken into account for withdrawing individuals from the study: interruption in follow-up visits in the control group and non-participation in the intervention group sessions and psychiatric admission. The CONSORT flowchart of the study design is shown in Figure 1.

### 2.3. Instruments

#### 2.3.1. Insomnia Severity Index

Insomnia severity was assessed using the Insomnia Severity Index (ISI) [35], a self-applied instrument designed to assess the severity of insomnia in the general population based on the diagnostic criteria of the DSM-IV and ICSD. A 5-point Likert scale was used to rate each item (0 = no problem; 4 = very severe problem), yielding a total score ranging from 0 to 28. The total score is interpreted as follows: not clinically significant insomnia (0–7); subthreshold insomnia (8–14); moderate insomnia (15–21); and severe insomnia (22–28) This scale has been found to have adequate psychometric properties in studies conducted using the English version [36], with internal reliability values (Cronbach’s a) between 0.74 and 0.90, and test–retest reliability equal to 0.89 one month after evaluation, 0.77 two months after, and 0.73 three months after. The Spanish version of the ISI scale was validated by Fernández-Mendoza et al. and Sierra et al. [37,38]

#### 2.3.2. Pittsburgh Sleep Quality Index

Sleep quality was analysed using the Pittsburgh Sleep Quality Index [39], which consists of 19 self-administered and self-rated questions. The 19 self-rated questions assess a wide variety of factors relating to sleep quality, including estimates of sleep duration and latency and the frequency and severity of specific sleep-related problems. These 19 items were grouped into 7 component scores. Each score weighted equally on a 0–3 scale. The seven component scores are then summed to yield a global PSQI score, which ranges from 0 to 21; higher scores indicate worse sleep quality [39]. This tool has shown acceptable measures of internal homogeneity, consistency, and validity in the English version. A global PSQI score greater than 5 yielded a diagnostic sensitivity of 89.6% and specificity of 86.5% in distinguishing between good and poor sleepers [39] and showed strong reliability and validity and moderate structural validity in a variety of samples, suggesting the tool fulfils its intended utility [40], as adapted in its Spanish version [41,42].

#### 2.3.3. EuroQol-5D Scale

Health-related quality of life (HRQoL) was assessed using the EuroQol-5D Scale (EQ-5D) [43]. This is a self-applied scale that is quick and easy to administer, producing a multidimensional description of general health as well as a numerical health profile. The scale is made up of two parts: the EQ-5D descriptive system, which consists of five dimensions (mobility, self-care, usual activities, pain/discomfort, and anxiety/depression), each with three levels of severity (value 1 = no problems, value 2 = some problems, value 3 = severe problems) and the visual analogue scale (EQ-VAS) (value 0 = worst, value 100 = best imaginable health status). This scale was validated in Spain by Xavier Badia [44]. For psychometric properties, the scale presented a test–retest reliability between 0.86 and 0.90 [45] and a strong correlation with the SF-36 scale. The EQ-5D scale has also been shown to be valid for use in patients with schizophrenia [46].

### 2.4. Group Treatments

#### 2.4.1. Description of the Psychoeducational Group Intervention

The intervention consisted of 6 sessions, 1 per week, each lasting 90 min. The 20 participants were divided into 4 groups of 5 people each. The intervention was conducted by a nurse. The objectives of the 6 sessions are described in Table 1. The program included: 1. First group meeting and sleep and insomnia’s basic concepts. 2. Sleep hygiene. 3. Stimulus control and reducing time in bed. 4. Relaxation techniques. 5. Cognitive therapy. 6. Summary of previous sessions and group farewell. To improve the patient’s active role in the intervention, each session was accompanied by activities to be completed at home.

During the sessions, as a measure of adaptation to the target population, the following aspects were addressed: presence of ruminations and invasive thoughts prior to conciliation. This characteristic is described as a common aspect in patients with schizophrenic disorder by Morrison and Baker [47] and which showed greater presence in patients with insomnia [48]; it is described as presence of maladaptive ruminations that negatively influence sleep reconciliation, according to Maria Luca [49]. Chiu et al. [50] already conclude that psychological sleep interventions must address voices and other psychotic symptoms. With a greater incidence of negative symptoms of schizophrenic disorders, knowledge about the importance of physical activity takes on relevance in this population [51]. The lack of activity presented by many of the patients [52], and the high presence of sedentary activities influence less sleep pressure and the alteration of the homeostatic cycle. Linking the community and rehabilitation resources reduces social isolation [9,53]. Little motivation for change is a common limitation since the patient believes that change is not possible [27], affecting the potential of CBT-I therapy [54]. Finally, narratives were also provided, during the sessions, from the exploratory study on the perception of insomnia by patients with schizophrenic disorder [32].

Participants continued with their existing pharmacological treatment. The group interventions were conducted in the ambulatory mental health centre. The nurse who performed the intervention is a member of the Severe Mental Disorders program and has training in mental health and in carrying out cognitive-behavioural interventions, as well as extensive healthcare experience. The place was equipped with the necessary space and equipment to carry out the intervention.

#### 2.4.2. Description of the Control Group

The members of the control group received their regular outpatient follow-up and treatment (visits with their psychiatrist of reference and mental health nurse). No different patterns were established during the visits. During their routine follow-ups, each patient was asked about their mental health status, general health, day-to-day habits, and treatment adherence, and a modification of the pharmacological prescription was applied if the symptomatology required it.

### 2.5. Analysis

#### Statistical Analysis

An exploratory and descriptive analysis was conducted using frequency tables for categorical variables and descriptive statistics for the quantitative variables. For the latter, normality assumptions were verified (Shapiro–Wilk), as well as homoscedasticity (Levene test) and sphericity (Mauchly). For the qualitative or ordinal variables, the Chi-squared test was used for the inter-subject comparison, and the McNemar’s test for intrasubject variability and Cramer’s V were used to measure the effect size of the associations, which is interpreted as a measure of the relative (strength) of an association between two variables. The coefficient ranges from 0 to 1 (perfect association).

With regard to the quantitative variables, the ANOVA parametric method was used for repeated measures to contrast the differences for the scales, evaluating the inter-subject, intra-subject, and interaction effects. In addition, the Tukey post hoc test was utilised for pairwise comparisons once the equality of variances was confirmed. Effect size was evaluated using Partial Eta Squared (*η*_*p*_2) [55] with the following interpretation: *η*_*p*_2 = 0.01 indicates a small effect; *η*_*p*_2 = 0.06 indicates a medium effect; and *η*_*p*_2 = 0.14 or higher indicates a large effect [56]. Pearson’s correlation coefficient was also used for the quantitative variables, to assess the correspondence between different scales according to the levels of study. All analyses were performed using the SPSS v21 program.

## 3. Results

### 3.1. Patients Characteristics

A total of 40 patients were included in this study. The patients were randomised between the intervention group (*n* = 20) and the control group (*n* = 20). Both groups did not present any statistically significant differences at the beginning of the study (*p* > 0.05). Table 2 shows the sociodemographic characteristics of the control group and the intervention group. In relation to the analysis variables, no statistical differences were observed. Both groups present the same insomnia and quality-of-life problems, Severity Index Insomnia (*t*: 0.476; *p* = 0.637), Pittsburgh Sleep Quality Index (*t*: 0.082; *p* = 0.935), quality of life, EQ-5D-VAS scale (*t*: 0.396; *p* = 0.694), as well as the values of the ISI ordinal scale (χ^2^: 3.604; *p* = 0.165) at the beginning of the study. Table 3 shows the scales’ scores at the initial time point.

### 3.2. Intervention Results

#### 3.2.1. Insomnia Severity Index (ISI)

The obtained results concerning the ISI Scale value (Table 4) show significant differences regarding the insomnia severity index at the three points of measurement, both in the intra-subject effect factor (F: 29.29; *p* = 0.000), in the inter-group factor (F: 23.65; *p* = 0.000), and in the intersection, which is the interaction between the two, (F: 28.36; *p* = 0.000) The effect sizes obtained in the intra-subject factor are large (*η_p_*2: 0.427), as they also were for the inter-group (*η_p_*2: 0.384), which suggest an important influence of the group on the score obtained in the ISI variable. In the post hoc analysis, relevant differences are observed in all pairwise comparisons, except for the comparison between the 6-month and 9-month measurement time points (*p* = 0.841).

In all cases, it becomes apparent that the ISI values are higher at the initial point of measurement in comparison with the values obtained in the posterior 6 and 9 months, with a mean difference of 3.63 (CI 95%: 2.02–5.23) (*p* = 0.000) and 4.10 (CI 95%: 2.45–5.75) (*p* = 0.000), respectively. Additionally, significant differences are observed in the comparison between groups, the control group being the one which presents higher scores with a mean difference of 5.63 (CI 95%: 3.29–7.98) (*p* = 0.000).

In the follow-up analysis over time, it is shown that the ISI values for the control group remain stable while there is a decrease in the mean scores in the intervention group between the initial time point and the next 6 months. However, there is no significant additional decrease between the 6- and 9-month time points (*p* = 0.841) (Figure 2).

Regarding the ISI scale analysis results based on the severity group (Table 5), it was noted that at the initial time point no relevant statistical differences were observed. However, after the 6-month follow-up, highly significant results were obtained (*p* = 0.009), with a large-scale effect (Cramer’s V: 0.541), clearly indicating that the intervention group was presenting insomnia cases of milder severity in comparison to the control group. At the initial time point, 50% (*n* = 10) of the sample presented with severe or moderate insomnia and at 6 months, only 10% (*n* = 2) presented with severe or moderate insomnia in the intervention group.

Additionally, after the 9-month follow-up, the effect was even more evident, as once again, highly significant differences were found with a considerable large effect size. In this case, the intervention group exhibited severity insomnia values considerably lower in comparison with the control group.

#### 3.2.2. Pittsburgh Scale

Regarding the PSQI variable (Table 6), the obtained results in the present study indicate that there are highly significant differences in the intra-subject’s factor (F: 17.88; *p* = 0.000), as well as in the inter-group factor (F: 13.73; *p* = 0.000) and in the interaction between the two (F: 18.31; *p* = 0.000), with large effect sizes in all three cases. In contrast, in the post hoc analysis, highly significant differences are observed between the initial measurement point and the 6- and 9-month time points, with a difference of 3.00 (CI 95%: 1.53–4.49) and 2.30 (CI 95%: 0.85–3.75), respectively. Nevertheless, no considerable distinctions at the measurement points of 6 and 9 months are observed. With regard to the comparison between groups, strongly significant differences are obtained, the control group being the one that presents higher scores, showing a mean difference of 3.77 (CI 95%: 1.71–5.83).

Overall, the results (Figure 3) suggest that the PSQI variable is sensitive to changes over time, showing relevant differences in both the within-subject and between-subject factors and the interaction between them. In addition, the results show that the control group presents poorer scores regarding the decrease in the PSQI variable than the intervention group, which suggests that the intervention is effective in improving sleep in this population.

#### 3.2.3. EQ-5D–VAS Scale Results

The results obtained for the EQ-VAS variable (Table 7) suggest that no significant differences are observed in the intra-subject factor (F: 0.240; *p* = 0.873); however, significant differences are obtained in the inter-group factor (F: 5.33; *p* = 0.27). In addition, highly significant differences are obtained in the interaction between the time of measurement and group membership (F: 12.05 *p* = 0.000), with a large effect size (*η_p_*2: 0.241). In the post hoc analysis by pairs, no significant differences are obtained in the comparison between the study points. However, differences are acquired in the comparison between the groups, with the mean score for the control group being lower in the intervention group in 10.48 (CI 95%: −19.66–−1.29) (*p* = 0.027).

Overall, the results indicate that the EQ-VAS variable does not present significant differences in the within-subject factor, but it does in the between-subject factor and in the interaction between the measurement point and group membership. Furthermore, important differences are observed between groups, the intervention group being the one with the highest scores in the EQ-VAS variable (Figure 4). These findings show that the intervention could have a positive impact on the perceived quality of life for the participants.

Regarding the analysis of the different EQ-5D scale variables, no statistical significance is obtained for any of the five dimensions (Table 8). In other words, it can be confirmed that no significant differences exist at the initial time point.

In the analysis of the acquired results at 6 months, statistically relevant differences were detected in the Self-Care (*p* = 0.003) and Pain (*p* = 0.16) variables. Regarding the Self-Care variable, a medium effect size is observed between both groups, with higher percentages of “No problems” in the intervention group (85%) compared to the control group (40%). The effect size demonstrated in the Pain variable also shows a medium effect, and in the same way as the Self-Care variable, the number of cases reported for “No problems” (70%) is significantly higher in the intervention group than in the control group (25%).

Finally, as presented in Table 8, at the 9-month time point, the only relevant differences for the Self-Care (*p* = 0.011) variable are obtained with a medium effect size (Cramer’s v: 0.402). The intervention group presents 3 out of 4 (75%) observations in the “No problems” response category, compared to a low 35% in the control group. The statistical significance observed in the 6-month time point for the Pain variable (*p* = 0.016) does not uphold in the 9-month time point (*p* = 0.053).

In the intra-group analysis performed using the McNemar test (Table 9), as it concerns a longitudinal study with categorical variables, statistically significant differences have been also observed.

Regarding the intervention group, relevant statistical differences have been regarded in the Pain variable (*p* = 0.002). It is observed how 10 patients progressed from “Some problems” at the pre-time point to “No problems” at 6 months. These differences persist at the 9-month time point for the intervention group in the Pain variable (*p* = 0.030), where a decrease in the severity is noticed. Seven observations progressed from “Some problems” to “No problems”, and even one observation changes from “Major problem” to “Some problems”, meaning there is a one level decrease in severity. However, there is one individual who experienced the opposite effect. While at the initial time point this person had “Some problems”, at the 9-month mark, they presented “Major problems”.

## 4. Discussion

Insomnia is a sleep disorder that clearly affects patients’ quality of life and worsens the prognosis of schizophrenia. Therefore, it is important to design and develop interventions that improve this sleep disorder. The treatment in this study has also been conducted in outpatient centres, demonstrating its effectiveness in a population with different comorbidities, both physical and psychiatric [57]. It has proven to be a cost-effective technique for improving insomnia’s symptomatology in the community setting [14,20], both in the general population and in patients with schizophrenic disorder [58]. Moreover, it can be performed by nurses, just as the previous pilot studies have shown [59,60,61], and has also been observed with the result of this pilot trial.

Nonetheless, it is important to adapt these interventions to the population with whom cognitive behavioural therapy will be conducted in order to provide a more accurately targeted therapy [62]. A number of studies have adapted and suggested adaptations for an improved intervention [30,31,63] and have obtained positive results in patients with a mental health disorder [22]. In our study, some of our adaptations are similar to those described by Waite et al. [31], including ruminations, poor sleep environment, lack of daytime activity, lack of evening activity, and night-time awakenings. Two previous studies [32,33] were conducted for this pilot trial to recognise and understand the patients’ experience of insomnia and the requirements and adjustments that it imposes on patients; thereby, this allowed the adaptation of the cognitive behavioural intervention for insomnia for patients with schizophrenic disorder. Including other patients’ narratives about their experience of insomnia and their profane strategies [32] has been interesting for the participants of the intervention group. According to Palagini et al. [64], insomnia’s treatment can play an important neuroprotective and preventative role. Moreover, CBT is a treatment with new and significant evidence of effectiveness for insomnia, psychopathology, and altered neuroplasticity rates.

The obtained results in our study are coherent with the ones observed in other studies regarding the efficacy of cognitive behavioural interventions, reducing insomnia’s severity and improving patients’ quality of life. Hwang et al. [23], in a sample of patients with schizophrenia, with an adapted intervention in which two fewer sessions were held and with a shorter duration of time per session noticed a decrease of 6.3 points in the ISI score in the intervention group at 8 weeks from the start. This improvement was also observed by Freeman et al. [24], who described a reduction in scoring in the ISI scale of 3.9 points in the control group and 9.3 points in the intervention group at the 12-week time point for evaluation; this improvement was also observed in patients with psychotic symptoms. In the CBT intervention performed by Nagai et al. [61], the decrease was −7.33 points (95% CI: −10.31 to −4.36) in the post-intervention assessment by nurses in the general population. 

In our study, the reduction in the score in the ISI scale was −6.9 (95% CI: −9.6 a −4.2) in the intervention group at the 6-month evaluation, reaching a reduction of −8.3 (IC 95%: −10.86 a −5.74) points at the 9-month mark. This improvement in the severity insomnia scale scores has also been described in the meta-analysis of Geiger-Brown et al. [57], where they reported an improvement of 9.49 points on the severity scale in a population with a mental disorder without discriminating by diagnosis, based on a total sample size of *n* = 566. Nevertheless, we have not observed any improvement in the control group’s scores following their usual follow-ups at the mental health centre.

Our intervention has also shown a reduction in the Pittsburgh Scale scores consistent with what has been observed in other studies, which have also shown that the cognitive behavioural intervention has been effective in improving insomnia [23,65]. Hawng et al. [23], in a 31 patient sample with schizophrenia, described an improvement of 5.8 points at the 8-week time point following the CBT intervention, 0.1 points greater than the improvement in our study at the 6-month time point, −5.7 (IC 95%: −7.9 a −3.5). Moreover, through a CBT intervention, Chiu et al. [65] investigated sleep subtypes in schizophrenia, and the response to cognitive behavioural therapy for insomnia observed an improvement in overall Pittsburgh scale scores by 4.1 points.

ISI and Pittsburgh, the scales used to evaluate the presence of insomnia and its severity, are diagnostic tools, which have proven utility in other studies [23,66,67]. The ISI Scale is an instrument that has been used in clinical practice for detecting and measuring the results of insomnia. Various studies show that ISI is a valid and reliable tool to evaluate insomnia and is sensitive in detecting treatment-related changes [36,68,69,70]. Yang et al. [71] recommend a reduction of 6 points to represent a clinically significant improvement in individuals with primary insomnia. In their study, Morin et al. [70] revealed that a decrease of >7 points in ISI was optimal for identifying participants with moderate improvements (60% sensitivity, 70% specificity), while a reduction of >8 in the ISI was optimal to identify participants with marked improvements (64% sensitivity, 80% specificity).

The reduction we detected in our intervention between the pre-intervention measure M = 15.00 ± 4.180 and at 9 months M = 6.70 ± 3.50 describes a difference of −8.3 (CI 95%: −10.86 a −5.74), this difference being statistically significant between both measures (*p* = 0.000). The effect size is also coherent with the one described by Nagai et al. [61], an intervention by nurses in the general population, who observed a high effect size in ISI scores from the initial time point to follow-up (Cohen’s d = 2.25). In our research, these data, analysed through eta squared, explain a large effect size in the intervention group analysis (F: 28.36; *p* = 0.000; *η_p_*2: 0.427).

On the other hand, the Pittsburgh scale has also been evaluated by Faulkner et al. [72] through qualitative analysis where evidence proves that patients with a schizophrenic disorder with <8 scores in this scale self-identify as good sleepers. This score was obtained on average by the intervention group in our research, with a post-intervention measurement of M = 7.30 ± 3.294.

Our CBT-I program has also shown positive aspects regarding health-related quality of life. The improvement in scale scores, which evaluate insomnia, has been accompanied by improvements in scores on the EQ-VAS scale. The intervention group progressed from a mean score of 54.75 ± 19.63 to an average score of 66.90 ± 14.01 (*p* = 0.006) at 9 months. The differences between the intervention and control groups have also shown statistically significant differences at 6 and 9 months. It is interesting to highlight the deterioration in quality of life within the control group at 6 and 9 months. Insomnia directly impacts quality of life, as proven in various studies, such as those conducted by Taylor et al. [73,74,75], who affirm that insomnia is associated with a worse general health state, negatively affecting one’s self-perception. Zeithlhofer et al. [76] also concluded the presence of a close connection between quality of life and insomnia. These differences were also detected in the prevalence of previous research among the schizophrenic population with and without insomnia [33].

If we delve into the analysis of quality of life based on the EQ-5D scale variables, which is a tool that has been used in various studies that evaluate insomnia [75,77,78] and has been utilised to measure the results of cognitive behavioural and psychoeducational interventions [19,61,79], we have observed that some variables improve significantly, but not all. These variables are Pain/Discomfort (*p* = 0.016) and Self-Care (*p* = 0.003). In the Enomoto et al. [80] meta-analysis, it was discovered that the improvement in insomnia through CBT-I is even more effective for pain management than specific cognitive behavioural therapy for chronic pain, an improvement that has also been detected in the studied population with mental disorders.

On the contrary, we have not observed improvements in the Anxiety/Depression variable; in addition, this is a facet of symptomatology that has not shown advancement in other studies [81], even though the CBT-I techniques have proven small to moderate upgrades in anxiety and depression [82]. The findings regarding quality of life have been consistent with the ones observed by Alimoradi et al. [83] in their review, affirming that CBT offers a potential effect, albeit small, on health-related quality of life.

### 4.1. Study Limitations

One of the strengths of our research is having conducted more appropriate statistical tests using a variety of scales, ISI and Pittsburgh, to obtain an answer for a smaller sample size. Another strong factor is that the study was performed in outpatient centres where patients carry out their follow-ups and monitoring. The randomisation of the sample and homogeneity between the control and intervention groups has also been a noteworthy aspect of our research.

Nonetheless, we cannot overlook that our study also presents some important limitations, like a reduced sample of participants, although this one was sufficient to conduct an initial pilot test. Another considerable limitation is not considering insomnia or schizophrenic disorders’ chronicity, nor have we considered other comorbidities that could have influenced our results. Another limitation is that we did not analyse the possible impact of our adaptation of the CBT-I on the intervention group individually and did not carry out a general analysis of the improvement of insomnia. The main objective of our intervention was the improvement of insomnia. The improvement of quality of life has only been analysed as a secondary objective, so the results obtained regarding health-related quality of life should be studied independently We also highlight as a limitation not analyzing the possible impact of our CBT-I adaptations on the intervention group individually and having carried out a general analysis of the improvement in insomnia. Finally, we did not track pharmacological patterns during the research period, although the scales that we used already include questions in this regard.

### 4.2. Future Implications and Future Lines of Research

The results obtained in this study prove that the proposed intervention is beneficial for improving insomnia in patients with schizophrenic disorder. They encourage us to continue focusing our interventions on this population group, which is not always included and does not always benefit from these insomnia treatment programs for the general population.

Our intention, once the positive results of our pilot study have been observed, is to consider expanding the sample of users to acquire greater and better evidence of the effectiveness of cognitive behavioural and psychoeducational therapy for treating this sleep disorder and improving patients’ quality of life with a mental disorder.

## 5. Conclusions

We can affirm that a cognitive behavioural intervention adjusted to populations with mental disorders, specifically schizophrenia, is effective in improving insomnia; thereby, the intervention leads to an enhancement in health-related quality of life. It is essential to include individuals with mental disorders in the design of these interventions, as the benefits are significant. Moreover, we can assert that a connection between insomnia and health-related quality of life exists, even in patients with mental disorders, and that this relationship varies based on the study group and the measurement time points. These findings suggest the importance of addressing sleep quality with patients in the management of the healthcare and the general well-being of individuals with mental health disorders.

## Figures and Tables

**Figure 1 jcm-12-06147-f001:**
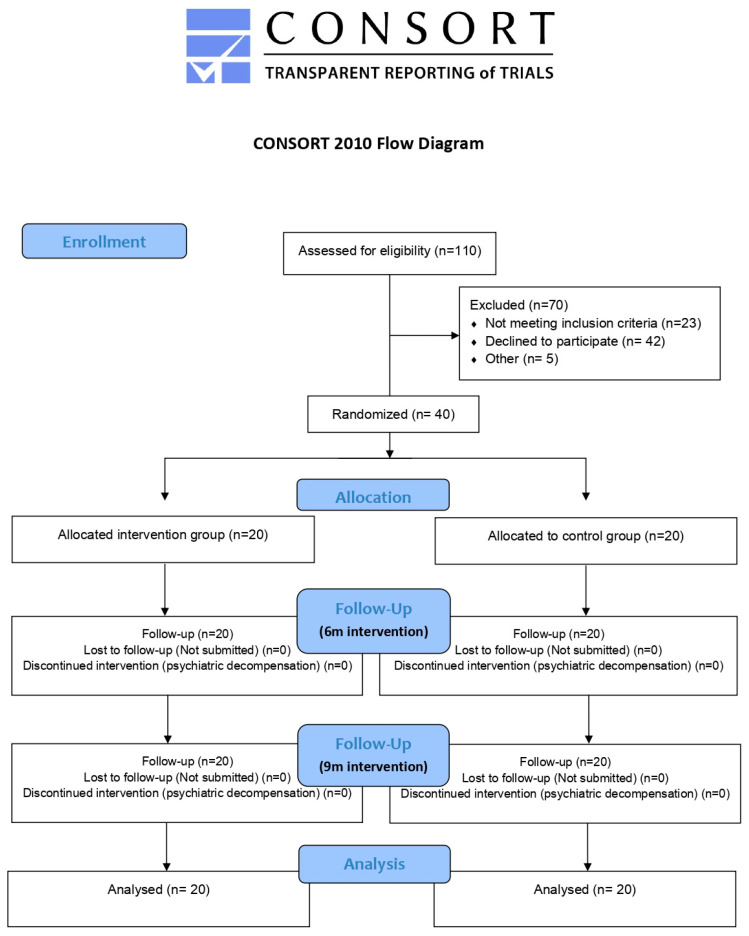
CONSORT flowchart of the study design.

**Figure 2 jcm-12-06147-f002:**
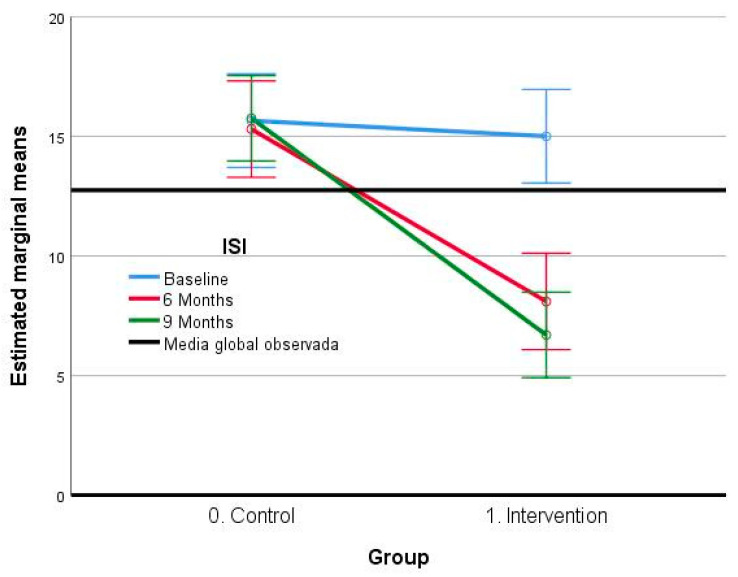
Estimated marginal means ISI scale.

**Figure 3 jcm-12-06147-f003:**
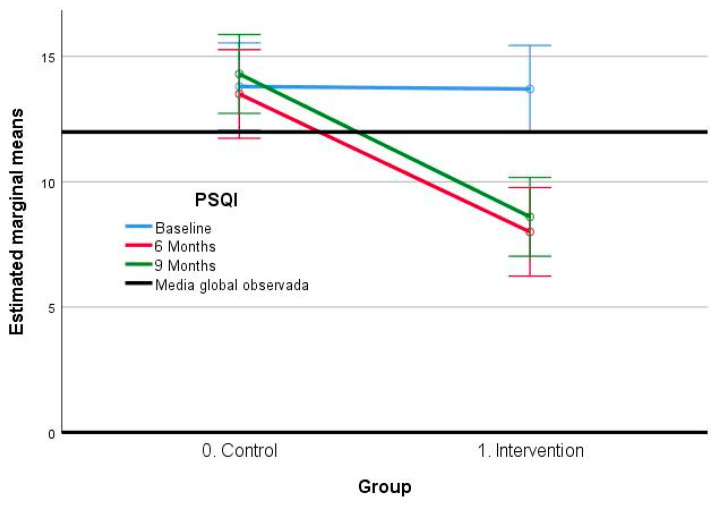
Estimated marginal means of the PSQI scale.

**Figure 4 jcm-12-06147-f004:**
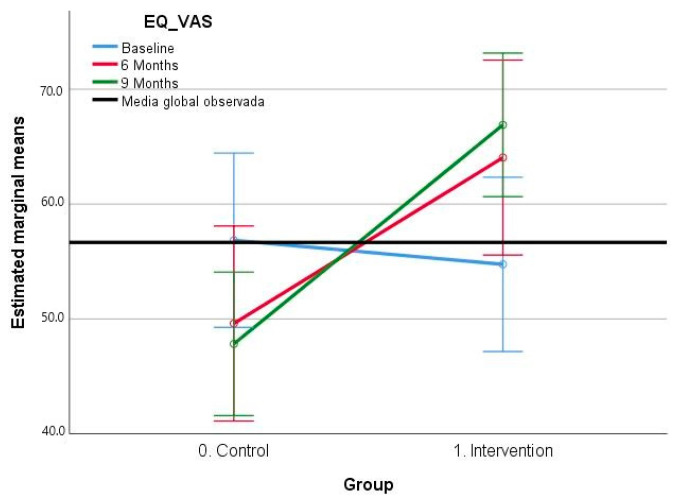
Estimated marginal means EQ-VAS scale.

**Table 1 jcm-12-06147-t001:** Psychoeducational and CBT program.

Session	Objective	Measure of Adaptation
1	First group meeting Sleep and insomnia’s basic concepts	Promoting CBT intervention potential Increasing motivation for change Patient narratives
2	Sleep hygiene	Increased physical activity Patient narratives
3	Stimulus control Reducing time in bed	Patient narratives
4	Relaxation techniques	Making mandalas Patient narratives
5	Cognitive therapy	Intervention for rumination control Patient narratives
6	Summary of previous sessions Group farewell	Encourage the linking of community and rehabilitative resources to reduce social isolation. Patient narratives

**Table 2 jcm-12-06147-t002:** Sociodemographic characteristics at the beginning of the study according to group.

	Control Group		Intervention Group		χ^2^/*t*	*p*
	*n*	%	*n*	%		
Sex					0.000	1
Male	12	60.0	12	60.0		
Female	8	40.0	8	40.0		
Age (mean ± SD)	50.35 (±12.75)		50.25 (±11.22)		0.026	0.979
Marital status					1.233	0.745
Single	13	65.0	11	55.0		
Married/In relationship	4	20.0	6	30.0		
Separated/Divorced	1	5.0	2	10.0		
Widowed	2	10.0	1	5.0		
Educational attainment					1.118	0.773
No primary school	1	5.0	1	5.0		
Primary school	9	45.0	8	40.0		
Secondary school	9	45.0	8	40.0		
University	1	5.0	3	15.0		
Employment status					1.467	0.690
Special work centre	0	0.0	0	0.0		
Freelancer	0	0.0	0	0.0		
Salaried employee	2	10.0	1	5.0		
Unemployed	1	5.0	0	0.00		
Disability	14	70.0	16	80.0		
Retired	3	15.0	3	15.0		
Degree of disability (mean ± SD)	65.74 (±4.86)		64.70 (±6.49)		0.562	0.577
Link to resources					1.382	0.710
No link	12	60.0	10	50.0		
SRC	2	10.0	2	10.0		
Pre-employment	3	15.0	2	10.0		
Social club	3	15.0	6	30.0		
Income level					1.048	0.592
No income	0	0.0	0	0.0		
Less than minimum wage	11	55.0	10	50.0		
Minimum wage	0	0.0	1	5.0		
More than minimum wage	9	45.0	9	45.0		
BMI (mean ± SD)	31.27 (±7.34)		32.06 (±6.38)		−0.365	0.717
Antipsychotics						
Yes	20	100.0	20	100.		
No	0	0.0	0	0.0		
Antidepressant					0.902	0.342
Yes	11	55.0	8	40.0		
No	9	45.0	12	60.0		
Mood stabiliser					1.558	0.212
Yes	5	25.0	2	10.0		
No	15	75.0	18	80.0		
Anxiolytics					0.102	0.749
Yes	11	55.0	12	60.0		
No	9	45.0	8	40.0		

**Table 3 jcm-12-06147-t003:** Scoring scales at the beginning of the study.

	Control Group		Intervention Group		χ^2^/*t*	*p*
	*n*	%	*n*	%		
Insomnia Severity Index Ordinal					3.604	0.165
No insomnia	0	0.0	0	0.0		
Mild insomnia	6	30.0	11	55.0		
Moderate insomnia	13	65.0	7	35.0		
Severe insomnia	1	5.0	2	10.0		
Index Severity Insomnia (mean ± SD)	15.65 (±4.45)		15.00 (±4.18)		0.476	0.637
Pittsburgh (mean ± SD)	13.80 (±4.31)		13.70 (± 3.29)		0.082	0.935
EQ-5D—Mobility					0.960	0.327
No problems	11	55.0	14	70.0		
Moderate problems	9	45.0	6	30.0		
Severe problems	0	0.0	0	0.0		
EQ-5D Self-Care					2.850	0.091
No problems	11	55.0	16	80.0		
Moderate problems	9	45.0	4	20.0		
Severe problems	0	0.0	0	0.0		
Usual activities					4.570	0.083
No problems	5	25.0	11	55.0		
Moderate problems	13	65.0	9	45.0		
Severe problems	2	10.0	0	0.0		
Pain/Discomfort					4.570	0.102
No problems	6	30.0	4	20.0		
Moderate problems	9	45.0	15	75.0		
Severe problems	5	25.0	1	5.0		
Anxiety/Depression					1.360	0.507
No problems	3	15.0	3	15.0		
Moderate problems	14	70.0	11	55.0		
Severe problems	3	15.0	6	30.0		
EQ-VAS (mean ± SD)	56.85 (±13.30)		54.75 (±19.63)		0.396	0.694

**Table 4 jcm-12-06147-t004:** ISI scale results.

	ISI	
	Control	Intervention
	M (SD)	M (SD)
Initial time point	15.65 (4.45)	15.00 (4.18)
6 months	15.30 (4.86)	8.10 (4.00)
9 months	15.75 (4.34)	6.70 (3.50)
	F	*η_p_*2
Intra-subject	29.29 ** (<0.000)	0.435 (Large)
Intersection	28.36 ** (<0.000)	0.427 (Large)
Inter-group	23.67 ** (<0.000)	0.384 (Large)
	MD	IC 95%
0 vs. 6 months	3.63 ** (<0.000)	2.02–5.23
0 vs. 9 months	4.10 ** (<0.000)	2.45–5.75
6 vs. 9 months	0.48 ^NS^ (0.841)	−0.61–1.56
Groups	5.63 ** (<0.000)	3.29–7.98

Note: ^NS^ Not Significant and ** Highly significant (*p* < 0.001). F: statistical value ANOVA test of one factor for paired samples; *η_p_*2: partial eta square (effect size); MD: mean difference; IC 95%: confidence interval at 95%.

**Table 5 jcm-12-06147-t005:** Contingency table and pairwise contrast test according to study groups. Variable: ISI (Insomnia Severity Index). Independent sample treatment. Moments: Initial time point, 6 months, 9 months.

	Initial Time Point		6 Months		9 Months	
	Control	Intervention	Control	Intervention	Control	Intervention
No Insomnia	0.0% (0)	0.0% (0)	15.0% (3)	50.0% (10)	0.0% (0)	55.0% (11)
Mild Insomnia	30.0% (6)	50.0% (10)	25.0% (5)	40.0% (8)	40.0% (8)	45.0% (9)
Moderate Insomnia	65.0% (13)	40.0% (8)	55.0% (11)	10.0% (2)	50.0% (10)	0.0% (0)
Severe Insomnia	5.0% (1)	10.0% (2)	5.0% (1)	0.0% (0)	10.0% (2)	0.0% (0)
	χ^2^ (*p*)	Cramer’s V	χ^2^ (*p*)	Cramer’s V	χ^2^ (*p*)	Cramer’s V
	3.60 (0.165 ^NS^)	0.300	11.69 (0.009 **)	0.541	23.06 (0.000 **)	0.759

Note: ^NS^ Not Significant and ** Highly significant (*p* < 0.001).

**Table 6 jcm-12-06147-t006:** PSQI scale results.

	PSQI	
	Control	Intervention
	M (SD)	M (SD)
Initial time point	13.80 (3.31)	13.70 (3.29)
6 months	13.50 (4.20)	8.00 (3.58)
9 months	14.30 (3.59)	8.60 (3.36)
	F	*η_p_*2
Intra-subject	17.87 ** (<0.000)	0.320 (Large)
Intersection	18.31 ** (<0.000)	0.325 (Large)
Inter-group	13.73 ** (0.001)	0.265 (Large)
	MD	IC 95%
0 vs. 6 months	3.00 ** (<0.000)	1.53–4.47
0 vs. 9 months	2.30 ** (<0.000)	0.85–3.75
6 vs. 9 months	−0.70 ^NS^ (0.240)	−1.67–0.27
Groups	3.77 ** (<0.000)	1.71–5.83

Note: ^NS^ Not Significantand ** Highly significant (*p* < 0.001). F: statistical value ANOVA test of one factor for paired samples; *η_p_*2: partial eta square (effect size); MD: mean difference; IC 95%: confidence interval at 95%.

**Table 7 jcm-12-06147-t007:** EQ-VAS scale results.

	EQ-VAS	
	Control	Intervention
	M (SD)	M (SD)
Initial time point	56.85 (13.30)	54.75 (19.63)
6 Months	49.60 (16.50)	64.05 (20.78)
9 Months	47.83 (13.60)	66.90 (14.01)
	F	*η_p_*2
Intra-subject	0.24 ^NS^ (0.783)	0.006 (Small)
Intersection	12.05 ** (<0.000)	0.241 (Large)
Inter-group	5.33 * (0.027)	0.123 (Medium)
	MD	IC 95%
0 vs. 6 months	−1.03 ^NS^ (0.999)	−7.04–4.99
0 vs. 9 months	−1.56 ^NS^ (0.999)	−7.55–4.42
6 vs. 9 months	−0.54 ^NS^ (0.999)	−5.52–4.44
Groups	−10.48 * (0.027)	−19.66–−1.29

Note: ^NS^ Not Significant, * Significant (*p* < 0.05), and ** Highly significant (*p* < 0.001). F: statistical value ANOVA test of one factor for paired samples; *η_p_*2: partial eta square (effect size); MD: mean difference; IC 95%: confidence interval at 95%.

**Table 8 jcm-12-06147-t008:** Contingency table. Summary table for variable EQ-5D (Mobility, Self-Care, Usual activities, Pain/discomfort, and Anxiety/depression) according to time points (Initial time point, 6 months and 9 months).

EQ-5D Mobility	Initial Time point		6 Months		9 Months	
	Control	Intervention	Control	Intervention	Control	Intervention
No problems	55.0% (11)	70.0% (14)	60.0% (12)	70.0% (14)	60.0% (12)	80.0% (16)
Some problems	45.0% (9)	30.0% (6)	40.0% (8)	30.0% (6)	40.0% (8)	20.0% (4)
Major problems	-	-	-	-	-	-
	χ^2^ (*p*)	Cramer’s V	χ^2^ (*p*)	Cramer’s V	χ^2^ (*p*)	Cramer’s V
	0.96 (0.327 ^NS^)	0.155	0.44 (0.507 ^NS^)	0.105	1.91 (0.168 ^NS^)	0.218
EQ-5D Self-Care	Initial time point		6 months		9 months	
	Control	Intervention	Control	Intervention	Control	Intervention
No problems	55.0% (11)	80.0% (16)	40.0% (8)	85.0% (17)	35.0% (7)	75.0% (15)
Some problems	45.0% (9)	20.0% (4)	60.0% (12)	15.0% (3)	65.0% (13)	25.0% (5)
Major problems	-	-	-	-	-	-
	χ^2^ (*p*)	Cramer’s V	χ^2^ (*p*)	Cramer’s V	χ^2^ (*p*)	Cramer’s V
	2.85 (0.091 ^NS^)	0.267	8.64 (0.003 **)	0.465	6.47 (0.011 *)	0.402
EQ-5D Usual activities	Initial time point		6 months		9 months	
	Control	Intervention	Control	Intervention	Control	Intervention
No problems	25.0% (5)	55.0% (11)	40.0% (8)	60.0% (12)	35.0% (7)	55.0% (11)
Some problems	65.0% (13)	45.0% (9)	60.0% (12)	40.0% (8)	50.0% (10)	45.0% (9)
Major problems	10.0% (2)	-	-	-	15.0% (3)	-
	χ^2^ (*p*)	Cramer’s V	χ^2^ (*p*)	Cramer’s V	χ^2^ (*p*)	Cramer’s V
	4.98 (0.083 ^NS^)	0.353	1.60 (0.206 ^NS^)	0.200	3.94 (0.139 ^NS^)	0.314
EQ-5D Pain/discomfort	Initial time point		6 months		9 months	
	Control	Intervention	Control	Intervention	Control	Intervention
No problems	30.0% (6)	20.0% (4)	25.0% (5)	70.0% (14)	25.0% (5)	55.0% (11)
Some problems	45.0% (9)	75.0% (15)	55.0% (11)	25.0% (5)	45.0% (9)	40.0% (8)
Major problems	25.0% (5)	5.0% (1)	20.0% (4)	5.0% (1)	30.0% (6)	5.0% (1)
	χ^2^ (*p*)	Cramer’s V	χ^2^ (*p*)	Cramer’s V	χ^2^ (*p*)	Cramer’s V
	4.57 (0.102 ^NS^)	0.338	8.31 (0.016 *)	0.456	5.88 (0.053 ^NS^)	0.383
EQ-5D Anxiety/depression	Initial time point		6 months		9 months	
	Control	Intervention	Control	Intervention	Control	Intervention
No problems	15.0% (3)	15.0% (3)	5.0% (1)	25.0% (5)	5.0% (1)	25.0% (5)
Some problems	70.0% (14)	55.0% (11)	70.0% (14)	60.0% (12)	80.0% (16)	60.0% (12)
Major problems	15.0% (3)	30.0% (6)	25.0% (5)	15.0% (3)	15.0% (3)	15.0% (3)
	χ^2^ (*p*)	Cramer’s V	χ^2^ (*p*)	Cramer’s V	χ^2^ (*p*)	Cramer’s V
	1.36 (0.507^NS^)	0.184	3.32 (0.190 ^NS^)	0.288	3.24 (0.198 ^NS^)	0.285

Note. ^NS^ Not Significant, * Significant (*p* < 0.05), and ** Highly significant (*p* < 0.001).

**Table 9 jcm-12-06147-t009:** Contingency Table and Pairwise Comparison Test according to study groups. Variable EQ-5D (Mobility, Self-Care, Usual activities, Pain/discomfort, and Anxiety/depression). Independent samples treatment.

EQ-5D Mobility	Control Group			Intervention Group		
	Initial Time Point	6 Months	9 Months	Initial Time Point	6 Months	9 Months
No problems	55.0% (11)	60.0% (12)	60.0% (12)	70.0% (14)	70.0% (14)	80.0% (16)
Some problems	45.0% (9)	40.0% (8)	40.0% (8)	30.0% (6)	30.0% (6)	20.0% (4)
Major problems	-	-	-	-	-	-
	Initial time point–6 months	Initial time point–9 months	6 months–9 months	Initial time point–6 months	Initial time point–9 months	6 months–9 months
Mc Nemar (*p*)	- (0.999 ^NS^)	- (0.999 ^NS^)	- (0.999 ^NS^)	- (0.999 ^NS^)	- (0.687 ^NS^)	- (0.500 ^NS^)
EQ-5D Self-Care	Control Group			Intervention Group		
	Initial time point	6 Months	9 Months	Initial time point	6 Months	9 Months
No problems	55.0% (11)	40.0% (8)	35.0% (7)	80.0% (16)	85.0% (17)	75.0% (15)
Some problems	45.0% (9)	60.0% (12)	65.0% (13)	20.0% (4)	15.0% (3)	25.0% (5)
Major problems	-	-	-	-	-	-
	Initial time point–6 months	Initial time point–9 months	6 months–9 months	Initial time point–6 months	Initial time point–9 months	6 months–9 months
Mc Nemar (*p*)	- (0.453 ^NS^)	- (0.289 ^NS^)	- (0.999 ^NS^)	- (0.999 ^NS^)	- (0.999 ^NS^)	- (0.500 ^NS^)
EQ-5D Usual activities	Control Group			Intervention Group		
	Initial time point	6 Months	9 Months	Initial time point	6 Months	9 Months
No problems	25.0% (5)	40.0% (8)	35.0% (7)	55.0% (11)	60.0% (12)	55.0% (11)
Some problems	65.0% (13)	60.0% (12)	50.0% (10)	45.0% (9)	40.0% (8)	45.0% (9)
Major problems	10.0% (2)	-	15.0% (3)	-	-	-
	Initial time point–6 months	Initial time point–9 months	6 months–9 months	Initial time point–6 months	Initial time point–9 months	6 months–9 months
Mc Nemar (*p*)	- (0.999 ^NS^)	1.00 (0.607 ^NS^)	- (0.999 ^NS^)	- (0.999 ^NS^)	- (0.999 ^NS^)	- (0.999 ^NS^)
EQ-5D Pain/discomfort	Control Group			Intervention Group		
	Initial time point	6 Months	9 Months	Initial time point	6 Months	9 Months
No problems	30.0% (6)	25.0% (5)	25.0% (5)	20.0% (4)	70.0% (14)	55.0% (11)
Some problems	45.0% (9)	55.0% (11)	45.0% (9)	75.0% (15)	25.0% (5)	40.0% (8)
Major problems	25.0% (5)	20.0% (4)	30.0% (6)	5.0% (1)	5.0% (1)	5.0% (1)
	Initial time point–6 months	Initial time point–9 months	6 months–9 months	Initial time point–6 months	Initial time point–9 months	6 months–9 months
Mc Nemar (*p*)	0.48 (0.788 ^NS^)	1.00 (0.801 ^NS^)	1.00 (0.607 ^NS^)	10.00 (0.002 **)	7.0 (0.030 *)	1.80 (0.407 ^NS^)
EQ-5D Anxiety/depression	Control Group			Intervention Group		
	Initial time point	6 Months	9 Months	Initial time point	6 Months	9 Meses
No problems	15.0% (3)	5.0% (1)	5.0% (1)	15.0% (3)	25.0% (5)	25.0% (5)
Some problems	70.0% (14)	70.0% (14)	80.0% (16)	55.0% (11)	60.0% (12)	60.0% (12)
Major problems	15.0% (3)	25.0% (5)	15.0% (3)	30.0% (6)	15.0% (3)	15.0% (3)
	Initial time point–6 months	Initial time point–9 months	6 months–9 months	Initial time point–6 months	Initial time point–9 months	6 months–9 months
Mc Nemar (*p*)	1.67 (0.435 ^NS^)	1.00 (0.607 ^NS^)	3.80 (0.284 ^NS^)	5.00 (0.082 ^NS^)	5.00 (0.082 ^NS^)	- (0.999 ^NS^)

Note. ^NS^ Not Significant, * Significant (*p* < 0.05), and ** Highly significant (*p* < 0.001).

## Data Availability

These study data are anonymised data. For original data, please contact the corresponding author; ethical approval does not cover making data openly accessible.

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
