# Peer review of "A Pilot Nurse-Administered CBT Intervention for Insomnia in Patients with Schizophrenic Disorder: A Randomized Clinical Effectiveness Trial"

_jcm, 2023, doi:10.3390/jcm12196147_

Round 1

Reviewer 1 Report

This article describes a group-based cognitive behavioral treatment for insomnia delivered to individuals receiving care for schizophrenia. While it is very worthwhile to examine sleep treatments in those with serious mental illness, there are significant concerns with the paper that reduced my initial enthusiasm. 

General comments: 

First, readability was significantly impaired by English language concerns. I would recommend the authors utilize editing resources for support in this regard. 

Second, it is unclear whether and how the authors adapted the intervention to this population. In my experience, those with such serious mental illnesses such as schizophrenia would experience significant barriers to participation in standard CBTI, namely disorganization of thought and behavior. How were those addressed, if at all? The discussion mentions adaptations generally, but I can find no mention of adaptations in the introduction or methods. If there were adaptations made, those must be clearly defined in order to evaluate the findings presented here. 

Thirdly, as mentioned in the limitations section, there were a lot of statistical analyses done "to give answer to a smaller sample size." This is opposite of the conventional wisdom that you must have adequate sample size for the number of analyses performed, as each additional comparison increases the probability of a false conclusion, so we must be judicious in the number of analyses performed in smaller subgroups. 

Specific Comments:

If this is a pilot study, it is important that this is described in the title, abstract, and introduction.

Introduction:

As there is a wide evidence base and many meta-analyses establishing the efficacy of cognitive behavioral therapy for insomnia (CBTI), it is not necessary to describe the history of it. Instead, I would recommend using that space to detail the evidence for CBT and CBTI treatments used specifically in schizophrenia, as this is an area with less research. What is the state of the literature of CBTI in serious mental illness? What studies have been done? How was CBTI adapted? What were their methods and what did they find? How might this study build on that evidence? 

Methods:

I would prefer to see a fuller description of how the participants for this study were recruited beyond a reference to another article. How were participants identified? Were they being seen for psychiatric or other behavioral treatments?

What were the credentials of those delivering the intervention? What did their training consist of?

It appears that the CONSORT diagram was not completed in its entirety (where it says "(give reasons)" the authors are required to describe the reasons for being discontinuation, for instance.

In the description of the psychoeducational group intervention, the program described does not match what is in the table and do not include any of the core components of CBTI. This is very confusing. Additionally, there is no description of how each component would be communicated or adapted to the schizophrenic population. Further description of session content would be really important here. 

Results:

In addition to my comment above about statistical analyses, I would also add that I do not see the added utility of performing correlations between measures. These are measures that we would expect would be pretty highly correlated. I do not quite understand why these correlations would increase statistical power. If the authors could provide a citation for their rationale, that would be helpful. 

Discussion:

In line with my comments on the introduction, I would limit the references to other studies to focus mostly on studies of CBTI in serious mental illness, as this population experience significant challenges that impact things like quality of life that make them less easily compared to the general population. Additionally, I would hesitate to discuss the effectiveness of CBTI for improving pain, as this is not the population of interest here. 

Here again I would further emphasize that it is hard to interpret these results as it is unclear what the intervention consisted of and if it was or was not adapted for the population of interest. 

As noted above, extensive editing will be required. There were significant issues with English syntax, grammar, and comprehension. There were many times where the words used were inappropriate for the context and impaired understandability. For instance, "baseline moment" or "initial moment" was used throughout. I am assuming that the word "timepoint" would be better word to use. 

Reviewer 2 Report

Manuscript ID jcm-2601361

Title: Nurse-administered CBT intervention for insomnia in patients with schizophrenic disorder: a randomized Clinical Effectiveness Trial

Abstract

The abstract is concise and well-structured, providing essential information.

Introduction

The introduction provides essential background information on the topic of sleep disorders and their relevance to individuals with schizophrenia. However, there are areas for improvement:

The introduction lacks smooth transition sentences between different ideas. Incorporate transitional phrases or sentences to connect related concepts and improve the overall flow of the text.

The Materials and Methods

This section provides a comprehensive description of the study's methodology, with some room for minor improvements in readability and visual aids.

Results

Regarding visual presentation, the figures presented are clear and contribute to the understanding of the results; however, consider labeling axes and data points more explicitly for improved clarity.

The Discussion and Conclusion

It effectively analyzed the study's findings, provide context through comparisons with previous research, acknowledge limitations, and propose relevant future directions.

No major revisions are needed for English.

Round 2

Reviewer 1 Report

First, I want to thank the authors for their close attention to detail and the quality of their edits. The edits made are clear and improve the manuscript significantly. 

I just have one final note. In the author's response letter (Point 9) they describe their previous studies and how it informed the current study. I would encourage the authors to find a way to integrate this information into the manuscript (I believe the end of the introduction would be a good place). 

For instance:

Modifications to CBTI were informed by... "(copy and pasted from author response) results obtained in the two previous studies. The study on the prevalence of insomnia and quality of life (Batalla‐martín et al., 2020) led us to focus in greater detail on aspects of health-related quality of life, due to its direct relationship with insomnia and with the perception of insomnia in patients with schizophrenic disorder, from our qualitative study (Batalla-Martín et al., 2022). In this study, patients described aspects such as periods of pre-conciliation rumination, lack of daytime activities, and detailed explanation of the effectiveness of CBT-I interventions instead of pharmacological treatments. Also, the sleep diary was used during all intervention weeks as a resource for participants to maintain the link between sessions. As stated by Mijnster et al (Mijnster et al., 2022), beyond the publication carried out by Waite et al (Waite et al., 2016), the available bibliography is limited to carry out adaptations. Therefore, our adaptation is based on our previous publications and on some evidence from the bibliography consulted."

Language is much improved. 
